# Inhibition of MEK1/2 Signaling Pathway Limits M2 Macrophage Polarization and Interferes in the Dental Socket Repair Process in Mice

**DOI:** 10.3390/biology14020107

**Published:** 2025-01-21

**Authors:** Angélica Cristina Fonseca, Priscila Maria Colavite, Michelle de Campos Soriani Azevedo, Daniela Carignatto Passadori, Jessica Lima Melchiades, Rafael Carneiro Ortiz, Camila Oliveira Rodini, Ana Paula Favaro Trombone, Gustavo Pompermaier Garlet

**Affiliations:** 1Department of Biological Sciences, Bauru School of Dentistry, University of São Paulo, Al. Octávio Pinheiro Brisola, 9-75, Bauru 17012-901, SP, Brazil; angelicafonseca@usp.br (A.C.F.); colavitepm@hotmail.com (P.M.C.); michelle_soriani@hotmail.com (M.d.C.S.A.); danicarignatto@gmail.com (D.C.P.); jessicalimamel1903@gmail.com (J.L.M.); rafaelortiz@usp.br (R.C.O.); carodini@usp.br (C.O.R.); 2Department of Health Sciences, Centro Universitário Sagrado Coração—UNISAGRADO, Bauru 17011-160, SP, Brazil; tromboneap@yahoo.com.br

**Keywords:** osteoimmunology, bone repair, MAP kinase signaling system

## Abstract

The bone-healing process involves the host inflammatory response, which in theory will mediate cell migration and the production of growth factors which will guide the repair process. In this scenario, macrophages are considered key elements in the development and resolution of inflammatory response. Interestingly, macrophages are highly responsive to microenvironment signals, which can guide their polarization into distinct functional phenotypes, denominated M1 (pro-inflammatory) and M2 (anti-inflammatory and pro-reparative). However, the factors responsible for macrophage polarization throughout bone repair remain unknown. In this study, we performed the pharmacological blockade of activated MEK1/2 kinases, intracellular elements theoretically involved in M2 polarization, to address their role in the dental socket repair process. After tooth extraction, our results demonstrate that the use of MEK1/2 inhibitor compound, namely PD0325901, limited the development of M2 response over time and resulted in an exacerbated non-resolving inflammatory response, which in turn impaired the bone repair outcome. Our results support the concept that chronic exacerbated inflammatory responses are not ideal for a proper bone repair outcome, pointing to M2 macrophages as essential elements for host response regulation and proper repair.

## 1. Introduction

Bone is a mineralized connective tissue composed of different cell types responsible for its formation, maintenance, and resorption. A continuous and dynamic bone remodeling contribute to bone physiological activities, such as the adaptation to mechanical stimuli and the storage/mobilization of mineral components, and also confers to the bone tissue a noteworthy healing capacity [1]. Bone repair involves a series of interconnected stages, including the development of transient inflammatory response, which in theory orchestrates the chemoattraction, activation, and differentiation of several cell types potentially involved in repair [2]. While the exact mechanisms connecting the inflammatory immune response with a favorable bone repair outcome remain unclear, the concept of “constructive inflammation”, i.e., a transient and controlled inflammatory response required for a proper bone repair outcome, is supported by the cause-and-effect association between anti-inflammatory drugs and delayed bone healing, as well as chronic unbalanced inflammation with impaired bone healing [3,4].

Among the inflammatory cells that allegedly contribute to bone repair, macrophages are considered as key elements of general tissue healing and are present in significant numbers in bone repair sites [5]. Noteworthily, macrophages are highly responsive to microenvironment signals, which can guide their polarization into distinct functional phenotypes, denominated M1 and M2 [6]. M1 macrophages are described as pro-inflammatory elements responsible for anti-microbial activity and extracellular matrix degradation, as well as increased osteoclastic activity [7], and are phenotypically characterized by the expression of CD80 and/or CD16/32 in mice and CD38 in humans [8]. While the classic elements associated with M1 polarization, namely PAMPs and Th1 cytokines, are not characteristically present in the sites of bone injury/repair, M1 macrophages predominate in the early stages of the bone repair process [9].

Conversely, M2 macrophages are classically described as elements that promote inflammatory process resolution and tissue repair and are identified mainly by the presence of the CD206 receptor [10]. M2 polarization can be mediated by Th2 cytokines, such as IL-4 and IL-13, not usually detected in bone injury/repair sites, similarly to M1-polarizing cytokines. Interestingly, despite the absence of classic M2-polarizing cytokines, M2 macrophages predominate in the late phases of the bone repair process [9]. Despite the nature of the extracellular signals that trigger macrophage polarization, intracellular signaling pathways, such as the RAS-RAF-MEK1/2-ERK1/2 pathway, are important for macrophage polarization towards an M2 profile [11]. In this scenario, the translocation of the intracellular signal by the phosphorylation of Janus kinase (JAK1) sequentially activates the signaling protein insulin receptor substrate-1 (IRS2), GRB2 adapter protein, ultimately inducing the activation of RAF-MEK1/2-ERK1/2 [12]. MEK1 (Map2k1) and MEK2 (Map2k2) are mitogen-activated protein kinases that activate the downstream effector molecules ERK1 and ERK2, directly involved in M2 differentiation [12]. Noteworthily, despite the absence of classic M2-polarizing cytokines such as IL-4, it is reasonable to suggest that other cytokines, such as VIP and PACAP, could mediate M2 polarization by targeting similar intracellular pathways [13,14].

At this point, it is mandatory to consider that despite the marked association of MEK1/2 inhibition with PD0325901-altered macrophage function, such a strategy may also impact other cell types relevant for the different stages of the bone repair process, considering the involvement of the MEK1/2 signaling in cells other than macrophages. For example, fibroblasts and other inflammatory/immune cells, cells that are present in different tissues participating in different experimental models such as malignant neoplastic models [15] or even inflammatory models of tissues such as lung [16], cardiovascular [17], or joint tissues [18]. Therefore, we emphasize that the model of choice for the study of the MEK1/2 pathway will be focused on macrophages present in intramembranous bone repair that evolves after tooth extraction.

Despite the uncertainness of the factors driving macrophage polarization in healing sites, the overall inflammatory immune response throughout the bone repair process appears to sequentially involve M1 and M2 macrophages, associated respectively with the local induction and resolution of inflammation that allow the development of the repair stages [19]. Therefore, M1 to M2 transition is regarded as an essential phenomenon, since a chronic non-resolving or exacerbated inflammatory response leads to impaired healing [4]. Accordingly, studies focused on favoring M2 polarization describe an increase in bone formation activity and a favorable healing outcome [13]. Similarly, endochondral ossification studies show that after tissue repair, we have the initial presence of M1 macrophages, being homeostatically replaced by or converted to M2 macrophages over time [20,21]. However, we must consider that such studies, in addition to being scarce, concern endochondral ossification, which differs from the intramembranous ossification that occurs in dental socket repair [20,21]. Therefore, considering that few associative studies focused on the potential role of macrophage subsets in bone healing [4,13], cause-and-effect studies are required to determine the real impact and the mechanisms underlying M1/M2 balance in bone-healing outcomes.

In this study, we propose an alternative approach to elucidate the role of macrophage polarization for the dental socket repair process, which, instead of using exogenous elements that can favor M2 polarization, takes advantage of compounds that interfere in the cell signaling process to block the development of the M2 phenotype. Specifically, this study’s experimental strategy comprises the pharmacological blockade of activated MEK1/2 kinases through the administration of the MEK1/2 inhibitor PD0325901 [22]. Indeed, although the factors involved in M2 polarization have not yet been identified in repair sites, given the general description of MEK1/2 involvement in macrophage polarization, the interference in this important signaling pathway can comprise a viable strategy to control macrophage phenotype and determine their role in the dental socket repair process.

Therefore, in this study, C57Bl/6 mice underwent the extraction of the upper right incisor and were treated (or not) with MEK1/2 inhibitor PD0325901. Then, they were comparatively evaluated regarding the intensity and nature of the inflammatory response and bone-healing development after tooth extraction and outcome by means of microtomographic, histological, birefringence, and molecular analysis in order to determine the role of MEK1/2 in M2 phenotype development in healing sites and its subsequent impact in bone-healing outcome.

## 2. Materials and Methods

### 2.1. Animals

The experimental groups consisted of 8-week-old male C57Bl/6 wild type (WT) mice acquired from Ribeirão Preto Medical School (FMR/USP) breeding facility, maintained during the experimental period in the facility of Department of Biological Sciences of FOB/USP. During the study period, the mice were fed with standard sterile solid mouse feed (Nuvital, Curitiba, PR, Brazil) and sterile water. The experimental protocol was approved by the Institutional Committee for Care and Animal Use and by the Guide for the Care and Use of Laboratory Animals (CEEPA-FOB/USP, process # 012/2019).

### 2.2. Experimental Protocol and Mice Tooth Extraction Model

Male C57BL/6 wild type (WT) mice were treated (MEK1/2i) or not (control groups). MEK1/2 inhibitor PD0325901 (Sigma Aldrich, San Luis, Missouri, USA—catalog number PZ0162-10 mg/kg IP, 24/24 h) [22], beginning 1 day prior to the upper right incisor extraction and throughout the experimental periods (0, 3, 7, and 14 days post tooth extraction). Mice were anesthetized by intramuscular administration of 80 mg/kg of ketamine chloride (Dopalen, Agribrans Brasil Ltd.a, São Paulo, Brasil) and 160 mg/kg of xylazine chloride (Anasedan, Agribrands Brasil Ltd.a) in the proportion 1:1 according to the animal body mass and the extraction of the upper right incisor was performed with the aid of a stereomicroscope (DF Vasconcellos S.A., Valença, Brasil) under 25× magnification as previously described [4,13,20].

The experimental sample size was 5 mice for each time point for each group (N = 5 group/time point) for microtomography (μCT), and then samples were prepared for histomorphometric, immunohistochemical, and collagen birefringence analysis, except for real-time PCR array in which 4 mice for each time for each group were used for experimental logistics reasons (N = 4 group/time point). At the end of the experimental periods, animals were euthanized with an excessive dose of anesthetic and the maxillae were collected. Samples for the μCT and histological analyses were fixed in PBS-buffered formalin (10%) solution (pH 7.4) for 48 h at room temperature, subsequently washed overnight in running water and maintained temporarily in alcohol fixative (70% hydrous ethanol) until the conclusion of the μCT analysis, and them decalcified in 4.13% EDTA (pH 7.2) and submitted to histological processing. Samples for molecular analysis were stored in RNAlater (Ambion, Austin, TX, USA) solutions [13].

### 2.3. Microcomputed Tomography (μCT) Assessment

The maxillae samples were scanned by the Skyscan 1174 System (Skyscan, Kontich, Belgium) at 50 kV, 800 μA, with a 0.5 mm aluminum filter and 15% beam-hardening correction, ring artifact reduction, 180 degrees of rotation, and an exposure range of 1 degree. Images were captured with 1304 × 1024 pixels and a resolution of 14 μm pixel size. Projection images were reconstructed using the NRecon V1.6.9.8 software and three-dimensional images were obtained by the CT-Vox 2.3 software. Morphological parameters of trabecular bone microarchitecture were assessed using the CTAn 1.1.4.1 software in accordance with the recommended guidelines. A cylindrical region of interest (ROI) with an axis length of 2.5 a 2.6 mm and diameter of 1 mm was determined by segmenting the trabecular bone located from the coronal to apical thirds. Trabecular measurements analyzed included the tissue volume (TV), bone volume (BV), bone volume fraction (BV/TV, %), trabecular thickness (Tb.Th, mm), trabecular number (Tb.N, mm), and trabecular separation (Tb.Sp) [2,23].

### 2.4. Histology Sample Preparation and Histomorphometric Analysis

Once scanned, the same specimens used in the uCT were prepared for histomorphometric analysis as previously described [4,13,20]. The samples were fixed and demineralized in EDTA pH 7.2 solution for an approximate period of forty days, with weekly changes of the demineralizing solution. After demineralization, the pieces were washed in distilled water and then submitted to the routine technique for embedding in Histosec^®^ paraffin (paraffin + resin). The paraffin blocks were cut into semi-serial histological sections in the transverse direction (perpendicular to the long axis of the dental socket). Then, 4 µm thick sections were obtained for staining with hematoxylin and eosin (HE), for staining with Picrosirius Red, as well as for immunohistochemical techniques, with intervals of 500 µm, covering the medial third of the dental socket.

Morphometric measurements were performed by a single investigator with a calibrated binocular light microscope (Olympus Optical Co., Tokyo, Japan) using a 100× immersion objective and a Zeiss kpl 8× eyepiece containing a Zeiss II integration grid (Carl Zeiss Jena GmbH, Jena, Germany) with 10 parallel lines and 100 points in a quadrangular area. The grid image was successively superimposed on approximately 13 histological fields per histological section, comprising all tooth sockets from the coronal limit adjacent to the gingival epithelium to the lower apical limit. For each animal/socket, sections from the coronal, medial, and apical thirds were evaluated. In the morphometric analysis, points were counted coinciding with the images of the following components of the dental socket: clot, inflammatory cells, blood vessels, fibroblasts, collagen fibers, bone matrix, osteoblasts, osteoclasts, and other components (empty space left by the inflammatory exudate or intercellular liquid and bone marrow) [20,24]. The results are presented as the volume density (mean) for each evaluated structure.

### 2.5. Collagen Birefringence Analysis

The Picrosirius polarization method and quantification of birefringent fibers were performed to assess the structural changes in the newly formed bone trabeculae matrix based on the birefringence of the collagen fiber bundles, as previously described [13,20]. All sections stained with Picrosirius Red stain were stained simultaneously to avoid variations due to possible differences in the staining process. Sections were analyzed through a polarizing lens coupled to a binocular inverted microscope (Leica DM IRB/E), and all images were captured with the same parameters (the same light intensity and angle of the polarizing lens was 90° to the light source). Adobe Photoshop CS6 software, version 13.0 Adobe, 2012 was used to delimit the region of interest (dental socket area comprising new tissue with the external limit comprising the dental socket wall), totaling 1,447,680 pixels^2^. The quantification of the intensity of birefringence brightness was performed using the AxioVision 4.8 software (CarlZeiss Microscopy GmbH, Jena, Germany). For quantification, the images were binarized for definition of the green, yellow, and red color spectrum, and the quantities of color pixels^2^ corresponding to the total area enclosed in the dental socket were measured. Mean values of 4 sections from each animal were calculated in pixels^2^.

### 2.6. Immunohistochemistry Analysis

We performed immunoexpression so that we could locate and quantify inflammatory cells (macrophages, granulocytes, and lymphocytes) present in the dental socket repair [20]. The material was pre-incubated with 3% Hydrogen Peroxidase Block (Spring Bioscience Corporation, Pleasanton, CA, USA) and subsequently incubated with 7% NFDM to block serum proteins. The histological sections from of all groups were incubated with Ly6G5B/Gr1 polyclonal antibody—sc168490 (Santa Cruz Biotechnology, Santa Cruz, CA, USA), F4/80 (SP115) monoclonal antibody—ab111101 (Abcam, Cambridge, UK), CD16 and CD32 (EPR22345) recombinant monoclonal antibody—ab228971 (Abcam, Cambridge, UK), CD206 (C-20) polyclonal antibody—sc34577 (Santa Cruz Biotechnology, Santa Cruz, CA, USA), and CD3-e (M-20) polyclonal antibody—sc1127 (Santa Cruz Biotechnology, Santa Cruz, CA, USA) at the manufacturer’s indicated concentrations for 1 h at room temperature. The identification of antigen–antibody reaction was performed using 3-3′-diaminobenzidine (DAB) and counter-staining with Mayer’s hematoxylin. Positive controls were performed in the mouse spleen. The analysis of immunolabeled cells was performed by a single investigator with a calibrated binocular light microscope (Olympus Optical Co., Tokyo, Japan) using a 100× immersion objective. For histomorphometric analysis, 13 histological regions were successively analyzed. The total number of immunolabeled cells was obtained to calculate the mean for each section.

### 2.7. Real-Time PCR Array Reactions

Real-time PCR arrays were performed as previously described. The extraction of total RNA from the remaining dental socket was performed with the RNeasyFFPE kit (Qiagen Inc, Valencia, CA, USA) according to the manufacturer’s instructions. The integrity of the RNA samples was verified by analyzing 1 mg of total RNA in a 2100Bioanalyzer (Agilent Technologies, Santa Clara, CA, USA) according to the manufacturer’s instructions, and the complementary DNA was synthesized using 3 μg of RNA through a reverse transcription reaction (Superscript III, Invitrogen Corporation, Carlsbad, CA, USA). Real-time PCR array was performed in a Viia7 instrument (LifeTechnologies, Carlsbad, CA, USA) using a custom panel containing targets “wound healing” (PAMM-121), “inflammatory cytokines and receptors” (PAMM-011), and “osteogenesis” (PAMM-026) (SABiosciences, Frederick, MD) for gene expression profiling. Real-time PCR array data were analyzed by the RT^2^ profiler PCR Array Data Analysis online software v3.5 (SABiosciences, Frederick, MD, USA). Gene expression normalization, which comprises an initial normalization with the initial geometric mean of three constitutive genes (GAPDH, ACTB, Hprt1) and subsequent normalization by the control group, was performed as previously described [13], and a subsequent normalization using a regular maxillary dental socket as the normalizing control was performed. Real-time PCR array analysis was conducted with samples from each experimental period (N = 4 animals/time/group), and the data are presented as the relative expression/fold change relative to the normalizing control (regular maxillary dental socket) as previously described [13].

### 2.8. Statistical Analysis

Differences among datasets were statistically analyzed by one-way analysis of variance (ANOVA) followed by the Tukey multiple comparison post-test or Student’s *t*-test where applicable; for data that did not fit in the distribution of normality, the Mann–Whitney and Kruskal–Wallis (followed by Dunn’s test) tests were used. The statistical significance of the experiment involving the PCR array was evaluated by the Mann–Whitney test, and the values were tested for correction by the Benjamini–Hochberg procedure [25]. Values of *p* < 0.05 were considered statistically significant. All statistical tests were performed with the GraphPad Prism 10.0 software (GraphPad Software Inc., San Diego, CA, USA).

## 3. Results

### 3.1. Expression of Different Molecules Present Between Groups Throughout the Evolution of Dental Socket Repair

We used a real-time PCR array for molecular analysis of gene expression patterns in dental socket repair of mice, comparing control groups with treated experimental groups (MEK1/2i) in the respective experimental periods of 3, 7, and 14 days (Figure 1). The results demonstrate a differential gene expression of several regulatory molecules of the dental socket repair microenvironment throughout its evolution, such as factors responsible for cell migration, formation of fibrous/blood/bone matrix, or even factors responsible for the reabsorption and remodeling of the aforementioned items [20]. Different molecules act to regulate the entire sequence of events in dental socket repair in control animals (C57Bl/6). Briefly, it is observed that in earlier periods of bone repair there are high levels of mRNA of growth factors (BMPs, TGFβ1, and VEGFa), extracellular matrix markers (COL1a1 and MMP1), mesenchymal stem cell markers (CD106, OCT-4, NANOG, CD34, CD105, and CD146), and cytokines and chemokines (IL-10 and TNF-α). Such molecules sequentially decrease their expression, giving way to markers with later expression peaks, such as extracellular matrix markers (COL1a2 and MMP2) and inflammatory components such as CCL2, CCL17, and CX3CL1. Some CCR2 and CCR5 chemokine receptors were upregulated over periods of 7 and 14 days. Furthermore, regarding bone regulators, it was observed that RUNX2 and ALPL were present early, with rapid positive regulation, while DMP1, PHEX, and SOST were expressed later in bone consolidation [20].

Among them, we highlight some growth factors (BMPs, TGFβ, VEGFs, and FGFs), extracellular matrix markers (COL1a1, COL1a2, and MMPs), bone markers (RUNX2, DMP1, ALPL, PHEX, SOST, and CTSK), chemokines and their receptors (CCL2, 3, 5, 12, 20, 22, CXCL1, 3, 12; CCR1, 2, 5; and CXCR1), mesenchymal stem cell markers (CD34, 105, 106, 116, 146, OCT-4, and NANOG), cytokines (IL-1B, 6, 10, TNF, and IFNG), and markers associated with macrophage polarization (iNOS, ARG, and FIZZ). Therefore, there was greater expression in the control group of BMP2 mRNA,4,7; IL-10; FIZZ; ARG; CCR2; CCR5; CCL2; CCL3; Col2a1; C106; NANOG; CD105; ALPL (*p* < 0.05 ControlXMEK1/2i at 7, 14 days); TGFb (*p* < 0.05 ControlXMEK1/2i at 7, 14 days); CCR1 (*p* < 0.05 ControlXMEK1/2i at 7, 14 days); CXCR1 (*p* < 0.05 ControlXMEK1/2i at 3, 7 days); CCL5; MMp1; CD116; OCT-4; RUNKL (*p* < 0.05 ControlXMEK1/2i at 7 days); CCL12; MMp2; DMP1; PHEX; SOST (*p* < 0.05 ControlXMEK1/2i at 14 days); CXCL12 (*p* < 0.05 ControlXMEK1/2i at 3, 7 days); Col1a2 (*p* < 0.05 ControlXMEK1/2i at 7, 14 days); MMp8 (*p* < 0.05 ControlXMEK1/2i at 7 days); CD34 (*p* < 0.05 ControlXMEK1/2i at 7 days); CD146 (*p* < 0.05 ControlXMEK1/2i at 3, 7 days); RUNX2 (*p* < 0.05 ControlXMEK1/2i at 3, 7, 14 days); CTSK (*p* < 0.05 ControlXMEK1/2i at 14 days) compared to the experimental groups. In contrast, there was lower expression in the control group of FGF1.2 mRNA (*p* < 0.05 ControlXMEK1/2i at 7, 14 days); VEGFb (*p* < 0.05 ControlXMEK1/2i at 7, 14 days); IL-1b; iNOS (*p* < 0.05 ControlXMEK1/2i at 7, 14 days); IL-6; CCL20 (*p* < 0.05 ControlXMEK1/2i in 7 days); TNF (*p* < 0.05 ControlXMEK1/2i at 3, 7, 14 days); CXCL1 (*p* < 0.05 ControlXMEK1/2i at 3 days) compared to the experimental groups.

No significative differences were observed regarding the gene expression patterns between the group treated with the drug diluent dimethyl sulfoxide and the untreated controls (Appendix A).

### 3.2. Immunolabeling Inflammatory Cells Between Groups Throughout the Evolution of Dental Socket Repair

#### 3.2.1. Granulocyte Ly6G5B and Lymphocyte CD3-e

To complete the analysis of the impact that the pharmacological blockade of the MEK1/2 pathways had on dental socket repair, we used immunolabeling for different types of inflammatory cells present there, thus, we initially used the markers Ly6G5B (GR1) and CD3-e (Figure 2) for granulocytes and T lymphocytes, respectively. At 0h we visualized the cells present in the blood clot; however, the two markers did not show statistical differences in this period when comparing the groups. At 3 days we observed a peak in the presence of granulocytes, other cells, as well as neutrophils that are the first to migrate towards repair. The control group had a higher absolute number of Ly6G5B+ cells (Figure 3A), followed by the MEK1/2i group with the lowest level of this marker. At 7 days, an equally important period for the migration and function of these inflammatory cells, we observed a decrease in immunolabeling in the control and MEK1/2i groups. In the final stage of inflammation, which is determinant for the evolution of repair and the main role of lymphocytes, CD3-e (Figure 3B) cells were observed with a peak at 14 days, once again in greater absolute quantity in the control group and less in the MEK1/2i group. Although the reported data are relevant, the groups have no statistical difference.

No significative differences were observed regarding the gene expression patterns between the group treated with the drug diluent dimethyl sulfoxide and the untreated controls (Appendix A).

#### 3.2.2. Macrophage CD206, F4/80, and CD16+CD32

In contrast, to visualize the direct effect of the pharmacological blockade of the MEK1/2 pathways, in macrophages we used immunostaining for the already consolidated markers of M0, M2, and M1, being anti-F4/80, anti-CD206, and anti-CD16+CD32 recombinants, respectively (Figure 2). In varying amounts, when present, these immunolabels follow the qualitative pattern described above. Cells, at 0 h, present within the blood clot or in the remnant of the periodontal ligament, or even in the granulation tissue, at 3 days and at 7 and 14 days in the remaining connective tissue. Thus, the number of CD206+ cells increased and was greater at 7 and 14 days, with a tendency of a lower absolute number of labeled cells in the MEK1/2i group, but without a statistical difference. For the quantification of F4/80+ cells decreasing with a peak at 3 days (in the control group) and 7 days (in the MEK1/2i group), we observed in all experimental periods the lowest number of markers for F4/80 (M0) in the MEK1/2i group, with no statistical difference. Markers for CD16&CD32+ (M1) showed a decrease over the periods and a peak at 3 days, although there was no statistical difference, and the MEK1/2i group showed a lower number of markers than the control group.

No significative differences were observed regarding the gene expression patterns between the group treated with the drug diluent dimethyl sulfoxide and the untreated controls (Appendix A).

### 3.3. Bone Microarchitecture Between Groups Throughout the Evolution of Dental Socket Repair

Through the µCT analysis we observed qualitative (Figure 4) and quantitative (Figure 5) data, such as the TV parameter (Figure 5F) showed no statistical difference within the same group and between different periods, as well as between different groups over the same periods, indicating the same distance from the long axis of the dental socket from coronal to apical points in all groups/periods. Thus, at 0 h (Figure 4), we observed the empty post-extraction tooth socket (hypodense) laterally delimited by the hyperdense cortices, seen from the cross-sectional planes and, mainly, from the cortical plane. A pattern like that described at 0 h is observed at 3 days (Figure 4), since in this period there is still no evidence of bone neoformation in all analyzed groups. On the 7th day (Figure 4) after tooth extraction, all groups showed a hyperdense structure starting from the cortical bone of the dental socket, indicating the presence of newly formed bone tissue. In the subsequent period up to 14 days (Figure 4), it is noted that the hyperdense structure is present in greater quantity, corresponding to the gradual filling of the interior of the dental socket. This understanding allowed us to observe the parameter of the density of the area occupied by bone volume—BV (Figure 5A)—which was lower in the MEK1/2i group (*p* < 0.05 at 14 days) when comparing it with the control. The pattern is similar in terms of the parameter of the proportion of bone volume per tissue volume—BV/TV (Figure 5B)—as well as the parameter Tb.Th (Figure 5C), which corresponds to the thickness of the newly formed bone trabeculae (*p* < 0.05 at 14 days). Results inverse to the BV parameters can be observed when analyzing the spaces present in the dental socket between the trabeculae formed, Tb.Sp (Figure 5E), with the control group having the smallest average space compared to the other groups both at 7 days and at 14 days (*p* < 0.05 at 14 days). Tb.N, the number of trabeculae (Figure 5D), shows that there is no statistical difference between the different groups.

No significative differences were observed regarding the gene expression patterns between the group treated with the drug diluent dimethyl sulfoxide and the untreated controls (Appendix A).

### 3.4. Density of Different Structures Present Between Groups Throughout the Evolution of Dental Socket Repair

Through the histomorphometric analysis, we observed that in the control group at 0 h (Figure 6), the dental socket was predominantly filled by a blood clot, which progressively organized itself as a framework for its invasion by constitutive cells, events that characterize the third day after tooth extraction in terms of the filling of the socket by inflammatory cells (ideally generating a low-intensity inflammatory process) and the presence of delicate collagen fibers, fibroblasts, and blood clots, which were gradually replaced by mature granulation tissue. On the seventh day (Figure 6) after tooth extraction, the dental alveoli were filled by a vascularized granulation tissue, with a greater predominance of mature collagen fibers and fibroblasts and a lower density of inflammatory infiltrate compared to the period described above. In addition, areas of new bone formation were also visible, starting from the dental socket cortex, where eventually we can see osteoclasts. At 14 days (Figure 6), the main event of dental socket repair was the predominance of primary bone tissue, permeated by a remnant of connective tissue rich in cells and blood vessels. It is worth mentioning that the primary trabeculae were in a clear remodeling process due to the presence of several trabecular surfaces occupied by active osteoclasts. In the MEK1/2i group, we initially observed the presence of blood clots for longer (7 days) and, mainly, lower bone density at 14 days (Figure 6).

When quantifying these events for better analysis, we observed the parameters related to the connective tissue (collagen fibers (Figure 7A); fibroblasts (Figure 7B); blood vessels (Figure 7C); and inflammatory infiltrate (Figure 7D)) and bone tissue parameters (bone matrix (Figure 7E); osteoblasts (Figure 7F); and osteoclasts (Figure 7G)) that can be quantified individually and together through the sum of the area densities of the structures corresponding to each of these tissues. In addition, other structures (Figure 7H), intracellular spaces, were also considered as a parameter. In the control group, the standard kinetics of evolution focused initially on the blood clot parameter (Figure 7I), and other structures, with a peak at 0 h, which decay over the subsequent periods with statistical difference in all groups (*p* < 0.05 0 h × 14 days). As for the collagen fiber (Figure 7A) and inflammatory cell (Figure 7D) parameters, this peak occurs at 3 days, with a subsequent decrease in the average over 7 and 14 days (*p* < 0.05 3 days × 0 h and 3 days × 14 days in all groups). At 7 days, the process of bone neoformation was in full swing with predominance of bone remodeling at 14 days (Figure 7E). Thus, although at 7 days there were several osteoblasts (Figure 7F), indicating a greater activity of synthesis, there was no statistical difference when compared with 14 days in the different groups, and the same was observed in relation to the bone matrix and blood vessel parameters (except in the MEK1/2i group for vessels).

At 0 h, as previously described, a predominance of blood clots (Figure 7I), microvessels, and surrounding fibrins (*p* < 0.05, ControlXMEK1/2i groups for vessels and fibers) was observed, as well as blank spaces (other structures—Figure 7H). There was no statistical difference between the groups in the same period in the clot and other structure parameters.

In the groups, a predominance of collagen fibers was observed (Figure 7A) at 3 days (*p* < 0.05). Two other important parameters in this period were blood clots (Figure 7I) and inflammatory cells (Figure 7D), the latter having no statistical difference in HE staining and the other having a difference when comparing the control with the respective MEK1/2i experimental groups (*p* < 0.05). It is worth mentioning that the parameters related to the bone tissue were not visualized at 3 days in the different groups because there was no neoformation with the presence of the already existing dental socket cortex, which was excluded in the histomorphometric quantification.

Subsequently, at 7 days, as mentioned, there was bone neoformation, thus, we observed a greater quantification of osteoblasts (Figure 7F) and bone matrix (Figure 7E), respectively, however, the parameters referring to connective tissue were still clearly present. Although the MEK1/2I group presented a lower density of area occupied by fibroblasts and blood clots, which determined a statistical difference from the control group (*p* < 0.05 at 7 days for fibroblasts), the presence of connective tissue permeating the newly formed bone tissue could still be observed. Regarding the bone matrix formed in this period, the MEK1/2i group has a lower average of bone matrix present compared to the other groups.

Thus, we observed the highest density of connective tissue in MEK1/2i, possibly determined by the greater amount of blood vessels (*p* < 0.05 MEK1/2iXControl), inflammatory cells, and fibroblasts in this group and in this period, with no statistical difference. Associated with this condition, the MEK1/2i group has, at 14 days, a higher density of area occupied by blood clots, without statistical difference. Bone parameters (bone matrix—Figure 7E—and osteoblasts—Figure 7F) showed higher density in control animals and lower density in MEK1/2i animals, with a statistical difference between these groups in both parameters (*p* < 0.05 MEK1/2iXControl for bone matrix and osteoblasts, at 14 days). The most relevant event in the period of 14 days was the peak of action of osteoclasts (Figure 7G), indicative of resorption and remodeling of bone tissue, which showed a lower density of osteoclasts in the MEK1/2i group (*p* < 0.05 MEK1/2iXControl).

The morphology of inflammatory cells, fibroblasts, osteoblasts and osteoclasts was considered for quantification (Appendix A). No significative differences were observed regarding the gene expression patterns between the group treated with the drug diluent dimethyl sulfoxide and the untreated controls (Appendix A).

### 3.5. Organization of Collagen Fibers Between Groups Throughout the Evolution of Dental Socket Repair

From the intensity of birefringence emitted by means of Picrosirius Red staining, we can assess the degree of maturation of the collagen fibers that formed the connective fibers and the matrix of newly formed bone trabeculae (Figure 8). The quantification of these fibers was in pixels^2^ (Figure 9A), total value, or even evaluated separately in different percentages of fibers with green, yellow, and red birefringence (Figure 9B). Thus, out of curiosity, we visualized at 0 h a remnant of periodontal ligament fibers, whose predominant birefringence was greenish due to the predominant constitution of thinner fibers (Figure 8). At 3 days, the pattern of visualization and quantification in absolute value was like that of the 0 h period, but without statistical difference. The dental socket was progressively filled by immature connective tissue, where at 7 days there was still a higher percentage of greenish fibers despite the presence of red and yellow fibers, with a difference between the control group and the experimental groups in absolute number (*p* < 0.05 ControlXMEK1/2i, at 7 days) (Figure 9A). At 14 days, these greenish fibers decreased and were replaced by red fibers composing the mature bone matrix. Although there is no statistical difference in the quantification at 14 days, we visualized that, in absolute numbers, the MEK1/2i group had a lower number of total fibers with a higher percentage of green fibers and a lower percentage of reddish fibers (Figure 9B).

No significative differences were observed regarding the gene expression patterns between the group treated with the drug diluent dimethyl sulfoxide and the untreated controls (Appendix A).

## 4. Discussion

Macrophages are considered key elements of the bone-healing process and consequently comprise a suitable therapeutic target for improving healing outcome [26]. Considering the alleged role of M2 macrophages in the late stage of the bone-healing process, in this study the pharmacological blockade intracellular MEK1/2 pathways was performed, with the administration of the MEK1/2 inhibitor PD0325901, in order to limit M2 response and consequently determine the role of this polarized macrophage subset in the dental socket repair process in a cause-and-effect way.

Initially, our results demonstrate that MEK1/2 inhibition effectively modulated M1/M2 balance in repair sites. While the untreated control group is characterized by an initial M1 dominance, followed by a switch towards predominant M2 type marker expression, which replicates previous studies [6,19], the MEK1/2i group presented an inversion in polarization pattern with a higher and sustained expression of prototypical M1 markers iNOS and TNF-α in parallel with a clear decrease in the expression of the M2 markers ARG1 and FIZZ. Noteworthily, the MEK1/2i group also presented a higher expression of pro-inflammatory cytokines and chemokines (IL-1b, IL-6, TNF-α, CCL2, and CCL5), reinforcing that MEK1/2 inhibition favors the predominance of M1 type response over time, resulting in a more pro-inflammatory environment [13,27]. However, the histomorphometric analysis does not demonstrate significant variations in the inflammatory cell count/density in the MEK/1/2i group. Also, the immunohistochemical analysis does not reveal significant changes in the counts of total macrophages (i.e., F4/80+ cells), M1 macrophages (CD16/32+), and M2 macrophages (CD206+), suggesting that the blockade of MEK1/2 may interfere partially in the M2 phenotype, perhaps limiting the cells’ function instead of the acquisition of the M2 phenotype throughout the repair process. Indeed, as previously mentioned, classic factors involved in M2 polarization such as IL-4 are absent in repair sites and, consequently, the exact extracellular signals and intracellular pathways involved in macrophage polarization in the repair process remain unclear, and therefore the inhibition of the MEK/1/2 pathway could be compensated by the other pathways [26,28,29]. In fact, the presence of distinct stimuli, such as IL-4 and/or IL-13, seems to modulate the effect of PD0325901 on the expression of repair-related genes in murine bone marrow-derived macrophages [26]. Accordingly, previous studies demonstrate that MEK1/2 inhibition can limit macrophage pro-inflammatory cytokine production without impairing other important functions [30] and can differentially modulate the expression of receptors and phenotypic markers [31], reinforcing the idea that distinct intracellular pathways can be responsible for specific functions and phenotype expression. This possibility is also supported by previous studies in the bone-healing context, where the immunoregulatory compound FTY720 favors bone formation, being associated with the modulation of M2 macrophages’ function but not due to interferences in its migration [32]. Despite the impossibility of fully mechanistically determining the effect of MEK1/2 inhibition on M2 commitment and function, the clear modulation of pro- and anti-inflammatory gene expression at healing sites supports the effective modulation of M1/M2 dynamics by the inhibitor PD0325901 and the subsequent analysis of the impact of such modulation on bone repair readouts. Indeed, different studies show that the immune/inflammatory response directly affects the tissue repair process [3,33], therefore the data described above directly correlate the inflammatory response with the delay and decrease in bone formation observed in the MEK1/2i group.

At this point, it is mandatory to consider that despite the marked association of MEK1/2 inhibition with PD0325901-altered macrophage function, this strategy may also impact other cell types relevant for the different stages of the bone repair process, considering the involvement of MEK1/2 signaling in cells other than macrophages. It is also important to mention that a significant modulation of neutrophils/granulocytes was observed, with decreased counts of GR1+ cells in the MEK1/2i group. Accordingly, previous studies demonstrate that PD0325901 suppresses neutrophil migration by interfering with the signaling required for adhesion and endothelial transmigration steps [34,35]. Accordingly, since PD0325901 can result in the reduction in neutrophil-mediated inflammation [30], the reduced neutrophil count in the healing sites of the MEK1/2i group can also account for the different healing outcomes observed in this group.

Specifically, impaired repair is characterized by a lower bone volume due to MEK1/2 inhibition, evidenced by µCT, in parallel with decreased bone and osteoblast density revealed by histological and histormophometric analysis and the decrease in collagen fiber area and maturity evidenced by the birefringence analysis. The molecular analysis provides interesting parallels with the tissue alterations observed. The MEK1/2i group showed higher expression of fibroblast growth factor (FGF1 and 2) [36] which correlates with the highest fibroblast density in the MEK1/2i group, mainly at 14 days. In this scenario, it is also important to consider the possible direct effects of PD0325901 on fibroblasts; however, considering that a previous study describes that PD0325901 exhibited concentration-dependent proliferation and growth inhibition of human conjunctival fibroblasts (HCFs) via cell-cycle arrest [37], it is possible to consider that the global modulation of the cytokine and growth factor milieu may have a more profound effect over fibroblasts’ fate and function throughout the repair process. According to the results of the present study, the MEK1/2i group presented a lower density of collagen fibers, and these fibers were more immature in the Picrosirius analysis, which is confirmed by the lower expression of matrix markers (COL1a2 and COL2a1) [36]. Picrosirius analysis also showed immaturity in the MEK1/2i group, with a reduction in red (organized) collagen fibers and an increase in green (disorganized) collagen fibers. These data are in line with a previous report that showed that, when inducing a tibial fracture, the administration of MEK1/2 inhibitor (PD0325901) altered endochondral ossification, both affecting the components of the matrix and its bone replacement, altering its bone architecture [38]. Regarding ECM-producing cells, there was a higher density of fibroblasts linked to a higher expression of fibroblast growth factor (FGF1 and 2) [39,40].

Another marker highlighted in the connective tissue parameter of the MEK1/2i group is the vascular endothelial growth factor (VEGF) which can also account for the higher density of blood vessels. These data are different in relation to the results obtained in a cancer model [39]. However, this increase in VEGF and vessels in the present article was the opposite of that previously found in a neoplasia model where M14 cells treated with the inhibitor PD0325901 resulted in a significant and dose-dependent inhibition of VEGF release [41]. Additionally, we observed a large amount of blood clots at all experimental times in the MEK1/2i group. Divergent results from ours show that the blockade of the MEK/ERK pathway by the inhibitor PD0325901, the same inhibitor used in this work, led to the inhibition of platelet activation, therefore, it is worth noting that although these differences may be related to several methodological aspects, both have a common point between them, that is, that the MEK1/2 pathway influences platelet activity [42].

The delay and decrease in bone formation parameters was confirmed by the lower expression of bone marker (RUNX2, ALPL, DMP1, PHEX, SOST RANKL, CTSK) [43] factors that interfere with osteoblast differentiation and activity, as well as osteoclast differentiation and bone resorptive capacity. Indeed, osteoclast counts/densities were also decreased in the MEK1/2i group. Noteworthily, an inverse pattern was observed with the increase in M2 activity by the administration of VIP and PACAP, which was correlated with increased bone marker formation and improved repair [13]. Regarding the decreased osteoblast counts/density observed upon MEK1/2 inhibition, the molecular analysis reveals a decrease in growth factors, such as BMPs, which could directly account for the interference with bone formation [13]. Additionally, the expression of MSC markers was also significantly decreased in the MEK1/2i group. Considering the key role of MSCs in the healing process, including bone repair, the decrease in their migration/activity can also account for the impaired repair due MEK1/2 inhibition [44]. At this point it is mandatory to consider that our results contrast with a previous description that PD0325901 improves Runx2 and alkaline phosphate gene expression in calvarial osteoblasts [38]. This divergence may be due to the isolated analysis of osteoblasts in culture in contrast with our in vivo analysis, where the concomitant effect of MEK1/2 inhibition in other cell types in the healing site, such as macrophages, may result in a final global effect distinct from the specific effect observed in vitro. Our results also differ from the report of improved tibial fracture repair in response to PD0325901, which could be due the distinct nature of the repair process, which involves cartilage formation [38]. Indeed, considering that MEK pathways can operate differentially in osteoblasts and chondrocytes, differential responses in intramembranous and endochondral bone formation can be expected [36,45]. While the distinct bone cell response observed in the MEK1/2i group may be result of the overall changes in the cytokine and growth factor milieu evidenced in the molecular analysis, it is important to consider that PD0325901 could also directly affect bone cells. Previous studies demonstrate that MEK1/2 inhibition results in a slight, but significant, increase in fracture union but did not promote substantive bone anabolism in the absence of an exogenous anabolic stimulus [45], suggesting that the direct effects of PD0325901 on osteoblasts in our model may be modest. Additionally, previous reports describe the inhibition of osteoclast formation by PD0325901 [46,47], suggesting that the decrease in osteoclast counts and in the expression of osteoclast activity markers observed in MEK1/2i can also be directly mediated by MEK1/2 inhibition. Furthermore, the presence of inflammatory mediators in the environment where MEK1/2 inhibition is evaluated seems to modulate the cell response outcome [46], reinforcing the hypothesis that in vivo evaluation with its inherent complexity is required for a more reliable estimation of the overall impact of MEK1/2 inhibition on the repair outcome.

At this point it is mandatory to consider and discuss some of the limitations of this study’s experimental design. It is important to discuss the route of administration of the inhibitor. The local application of the MEK1/2 inhibitor only in the region of the tooth socket after tooth extraction could lead to a lower and more effective dose of the drug, allowing us to have more evident results. However, while it is not possible to precisely determine the length of PD0325901’s effect after injection, considering the relatively long duration of the experimental model used in this study, it was assumed that repeated injections would be required to ensure the effect of the inhibitor at all the time points evaluated. Therefore, while it could be possible to locally deliver the inhibitor in the alveoli right after the tooth extraction, with the subsequent development of a bone clot, followed over time by the development of a granulation tissue and its replacement by neoformed bone tissue, the effectiveness of the local injection would be questionable, since it would require mechanical interference in the healing tissue, even considering the use of a fine needle, possibly representing a limitation to the dispersion of the injected compound.

Another relevant point refers to the possible in vitro analysis of PD0325901 in macrophages throughout their polarization process in a bone-healing environment. We initially clarify that previous studies [26,30,38] performed in vitro investigations of PD0325901’s effects on macrophage polarization, including a relatively recent investigation performing a broad characterization of macrophage polarization mechanisms and identification of M2 type polarization inhibitors [11], which included PD0325901, described in the study as one of the most potent inhibitors of IL-4-induced macrophage polarization in the screens in which multiple highly selective MEK inhibitors were tested. Additionally, a previous study demonstrates that PD0325901 significantly increased the expression of IL-4/IL-13 (M2)-responsive genes in murine bone-marrow-derived and dental socket macrophages. However, in vitro experiments focused on the M2 polarization process are essentially based in the use of IL-4 as the prototypic M2-polarizing factor, but as previously mentioned, Th2 cytokines, such as IL-4 and IL-13, are not usually detected in bone injury/repair sites. Therefore, while it is possible to suggest that other cytokines, such as VIP and PACAP, could mediate M2 polarization in healing sites [13], the use of IL-4-based experiments would replicate previous data, and the use of other factors would require broader experiments beyond the scope of this study. It is also essential to carefully interpret macrophage phenotypic data considering the limited extent of the markers used. While the immunohistochemical analysis does not evidence significant variations in the counts of alleged M1 and M2 macrophages, the molecular analysis of M1 and M2 markers suggests a modulation of M1/M2 balance. Therefore, the assumption of the M1/M2 balance modulation is based in the parallel analysis of both experiments, and definitive and stronger evidence regarding the exact nature of macrophages’ polarization dynamics throughout the bone repair process would require broader panels of phenotypic and functional markers.

Finally, it is necessary to consider that the present study proposes an alternative approach to elucidate the role of macrophage polarization for the dental socket repair process. Instead of using exogenous elements that can favor M2 polarization, we take advantage of compounds that interfere in the cell signaling process to block the development of the M2 phenotype. Therefore, despite the effectiveness of using the PD0325901 compound to demonstrate the effect of MEK1/2 inhibition on bone-healing outcomes in a cause-and-effect way, and the constructive role of M2 macrophages required for proper bone healing, this strategy would not make sense from the translational viewpoint since it limits the healing process. However, the results presented here can provide the basis for future studies with translational potential, perhaps focused in improving—and not inhibiting—MEK1/2 signaling. In this context, it is also important to consider that future strategies based in the modulation of MEK1/2 signaling also must consider a comprehensive analysis of the multiple cell types involved in the bone-healing process, beyond macrophage polarization.

## 5. Conclusions

We conclude that our results suggest that the inhibition of MEK1/2 plays an important role in dental socket repair, making it delayed in relation to the control, initially involving the control of the migration of inflammatory cells and subsequent bone formation.

## Figures and Tables

**Figure 1 biology-14-00107-f001:**
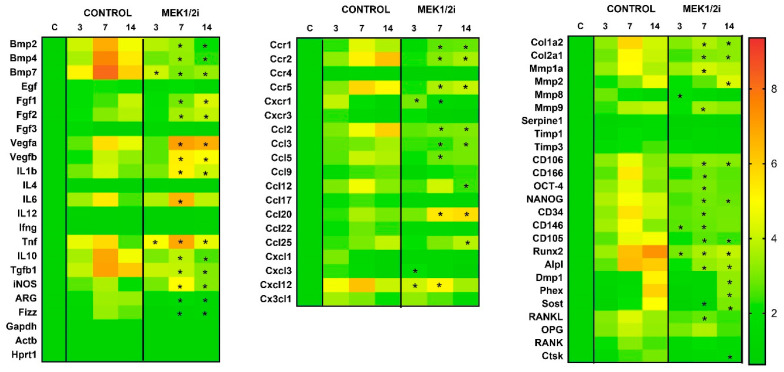
Comparative molecular analysis (PCR array) in the dental socket repair process over the periods of 0, 3, 7, and 14 days after tooth extraction between control and MEK1/2i groups. Heat map to quantify the expression of the growth factors (BMPs, TGFβ, VEGFs, and FGFs), extracellular matrix markers (COL1a1, COL1a2, MMPs, TIMPs, and Serpine), bone markers (RUNX2, DMP1, ALPL, PHEX, SOST, CTSK, RANKL, RANK, and OPG), chemokines and their receptors (CCLs, CXCLs, CCRs and CXCRs), mesenchymal stem cell markers (CD34, 105, 106, 116, 146, OCT-4, and NANOG), cytokines (ILs, TNF, and IFNG), and markers associated with macrophage polarization (iNOS, ARG, GAPDH, ACTb, HPRT-1, and FIZZ). * (*p* < 0.05) indicates statistically significant differences between the MEK1/2i and control groups.

**Figure 2 biology-14-00107-f002:**
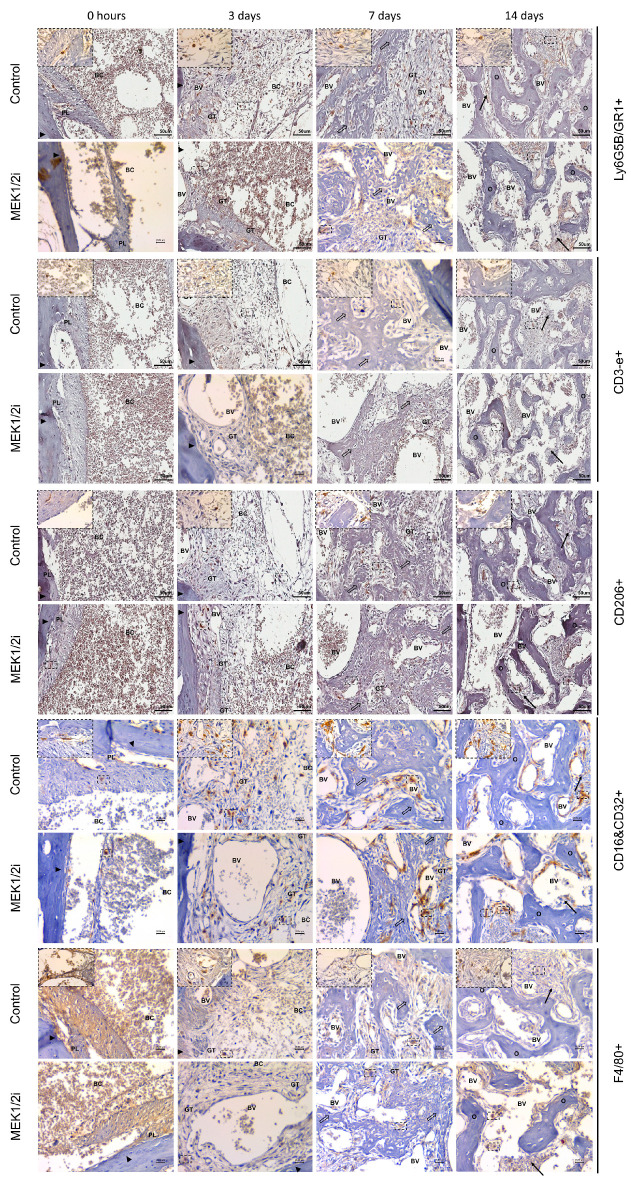
Comparative immunohistochemical analysis for anti-Ly6G5B/GR1, anti-CD3-e, anti-CD206, anti-CD16&CD32 and anti-F4/80 antibody in the dental socket repair process over the periods of 0, 3, 7, and 14 days after tooth extraction between control and MEK1/2i groups. Photomicrographs are representative of the middle region of the dental socket, where there is the presence of a blood clot (BC), blood vessels (BV) and preservation of the dental socket cortex (closed arrows) immediately after tooth extraction, with its progressive replacement by a granulation tissue (GT) and beginning of bone neoformation (open arrow). In later periods we see the persistence of connective tissue (thin arrow), bone trabeculae in remodeling (O), and the varied presence of + cells (square). IHC staining: Scanned in the Aperio Scanscope CS instrument with a 40× objective and under the Leica MC170 optical microscope with 40× (Bar = 50 μm) and 100× objectives (Bar = 20.34 μm).

**Figure 3 biology-14-00107-f003:**
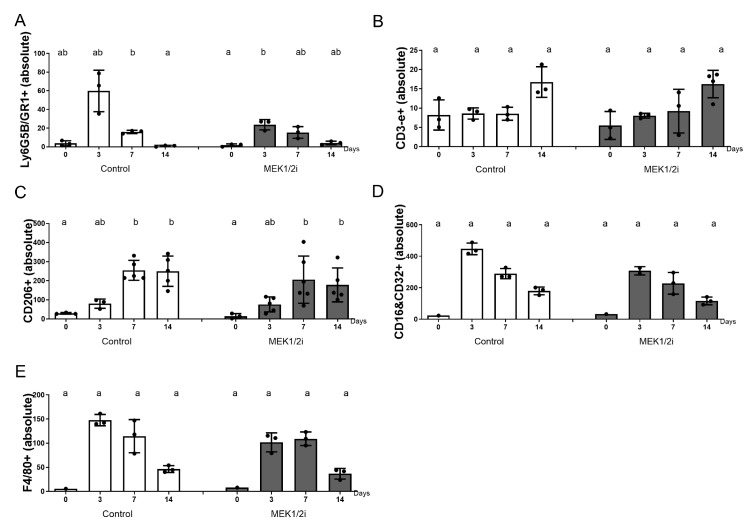
Comparative analysis of the quantification of (**A**) Ly6G5B/Gr1+, (**B**) CD3-e+, (**C**) CD206+, (**D**) CD16&CD32+, and (**E**) F4/80+ cells in absolute numbers. Different lowercase letters represent a statistically significant difference (*p* < 0.05) between different time points within the same group; time points within the same group presenting the same letter are not statistically different.

**Figure 4 biology-14-00107-f004:**
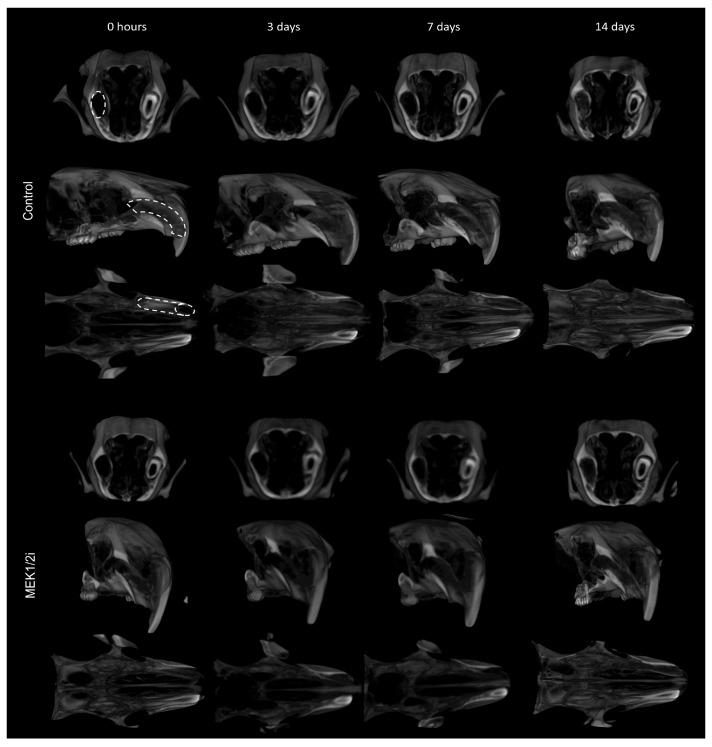
Three-dimensional morphological analysis by µCT of the dental socket repair process over the periods of 0, 3, 7, and 14 days after tooth extraction between control and MEK1/2i groups. The geometric figures drawn in the Control group represent the conformation of the dental sockets in different section planes. Coronal, sagittal, and transversal section planes are shown.

**Figure 5 biology-14-00107-f005:**
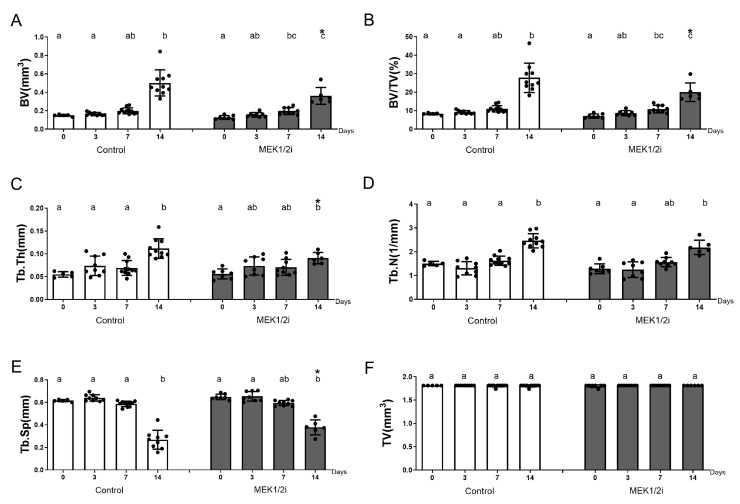
Comparative analysis of morphological parameters of bone trabecular microarchitecture in the dental socket repair process over the periods of 0, 3, 7, and 14 days after tooth extraction between control and MEK1/2i groups. Bone trabecular analyses (**A**–**F**) included: (**A**) Bone volume (BV), (**B**) bone fraction in relation to total volume (BV/TV), (**C**) trabecular thickness (Tb.Th), (**D**) number of trabeculae (Tb.N), (**E**) mean distance between trabeculae (Tb.Sp), (**F**) total tissue volume (TV). The results represent the mean and standard deviation values in each of the analyzed periods. Different lowercase letters represent a statistically significant difference (*p* < 0.05) between different time points within the same group; time points within the same group presenting the same letter are not statistically different. * (*p* < 0.05) indicates statistically significant differences between the MEK1/2i group versus control group at the specified time point.

**Figure 6 biology-14-00107-f006:**
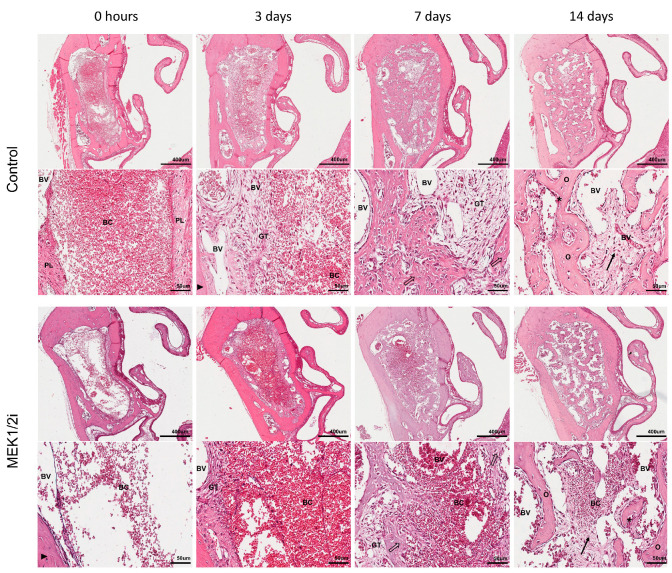
Comparative panoramic histological analysis in the dental socket repair process over the periods of 0, 3, 7, and 14 days after tooth extraction between control and MEK1/2i groups. Photomicrographs are representative of the middle region of the dental socket, showing the presence of blood clot (BC), blood vessels (BV) and preservation of the dental socket cortex (closed arrows) immediately after tooth extraction, its progressive replacement by a granulation tissue (GT), and beginning of bone neoformation (open arrow). In later periods we see the persistence of connective tissue (thin arrow), osteoclasts promoting resorption (*), and bone trabeculae in remodeling (O). HE staining: Scanned on the Aperio Scanscope CS device with 10× (Bar = 400 μm) and 40× objectives (Bar = 50 μm).

**Figure 7 biology-14-00107-f007:**
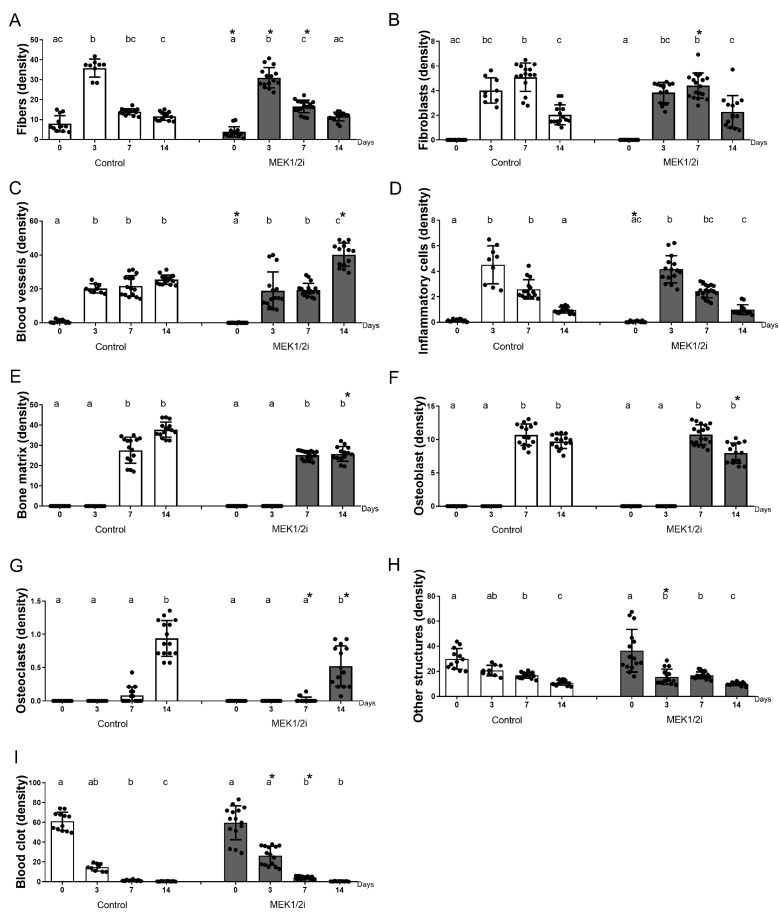
Comparative analysis of the total area density (%) occupied by (**A**) collagen fibers, (**B**) fibroblasts, (**C**) blood vessels, (**D**) inflammatory cells, (**E**) bone matrix, (**F**) osteoblasts, (**G**) osteoclasts, (**H**) other structures, (**I**) blood clot in the dental socket repair process over the periods of 0, 3, 7, and 14 days after tooth extraction between control and MEK1/2i groups. The results represent the mean and standard deviation values of the analyzed period. Different lowercase letters represent a statistically significant difference (*p* < 0.05) between different time points within the same group; time points within the same group presenting the same letter are not statistically different. * (*p* < 0.05) indicates statistically significant differences between the MEK1/2i and control group at the specified time point.

**Figure 8 biology-14-00107-f008:**
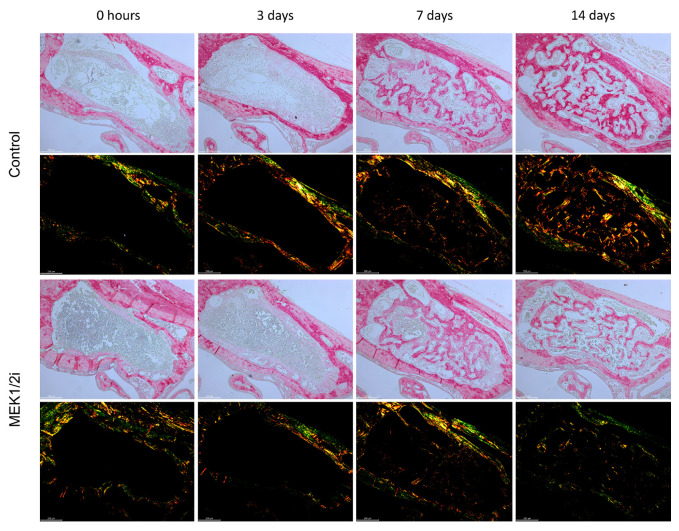
Comparative birefringence analysis of collagen fibers in the dental socket repair process over the periods of 0, 3, 7, and 14 days after tooth extraction between control and MEK1/2i groups. Photomicrographs are representative of the middle region of dental socket, captured under conventional light and polarized light. Green birefringence color indicates thin fibers; yellow and red colors in the birefringence analysis indicate thick collagen fibers. Picrosirius red coloration; objective of 10×; Bar = 200 μm.

**Figure 9 biology-14-00107-f009:**
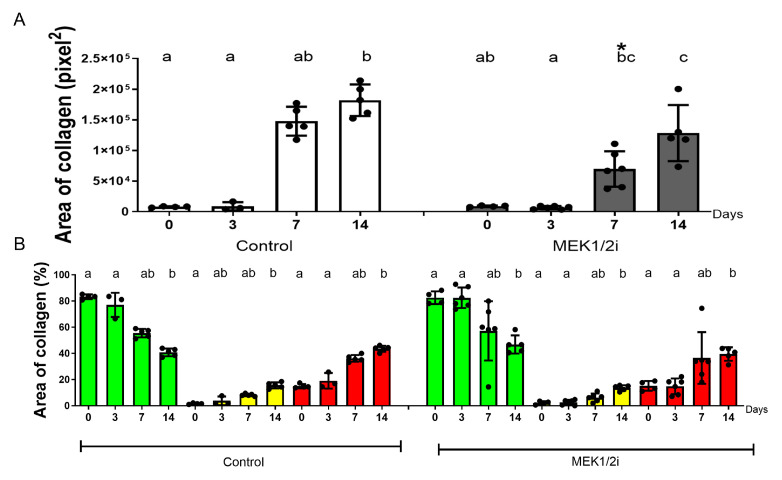
Comparative analysis of the total area of collagen fibers ((**A**)—pixel^2^) and the distinction in percentage of the degree of maturation of collagen fibers in green, yellow, red birefringence colors ((**B**)—%) in the dental socket repair process over the periods of 0, 3, 7, and 14 days after tooth extraction between control and MEK1/2i groups. Results are presented as the mean (±SEM) of percentage or pixels^2^ for each color in the birefringence. Different lowercase letters represent a statistically significant difference (*p* < 0.05) between different time points within the same group; time points within the same group presenting the same letter are not statistically different. * (*p* < 0.05) indicates statistically significant differences between the MEK1/2i and control group at the specified time point.

## Data Availability

The datasets generated and analyzed during this study are available from the corresponding author on reasonable request.

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
