# Peer review of "Inhibition of MEK1/2 Signaling Pathway Limits M2 Macrophage Polarization and Interferes in the Dental Socket Repair Process in Mice"

_biology, 2025, doi:10.3390/biology14020107_

Round 1
Reviewer 1 Report
Comments and Suggestions for Authors
In this manuscript, the author treated mice with a MEK inhibitor after tooth extraction and assessed how this treatment affected the polarization of macrophage and bone regeneration. Several standard techniques in the field, such as micro-CT, histomorphometry, and birefringence were utilized here. However, considering the novelty and scientific rigor of current manuscript, I do not think it is ready for publication. Below are my detailed comments.
1. There are many typos and grammar errors throughout the manuscript, making the manuscript very challenging and distracting to read. Just name a few, the title, line 14, “theory mediate the cells migration …”, line 23, “resulted in an exacerbated a non-resolving”, line 71, “macrophages o predominate in the early stages”, and many more. The current manuscript is lengthy and wordy, which further increases the difficulty in comprehending the main message the authors want to convey.
2. I didn’t see the novelty of this manuscript, given previous studies have shown the regulation of macrophage polarization and bone repair by MEK signaling, such as PMID: 31801908, 38352056. What new knowledge does this current study bring?
3. Figure1, what conclusion are the authors trying to draw?
4. For the figures, what does the labels of a, b mean? The statistical results are very confusing. It is not clear how many biological repeats are performed in each experiment.
5. For me, the representative images in fig. 4 are not as different as the authors claimed.
6. Is it Ly6c5b or ly6g5b (lines 309, 313)?
7. Both “M1” and “M2” cells are reduced in the drug treatment group. It is unclear which subtype contributes to the delayed bone recovery.
8. Related to the above, MEK inhibition brings many biological effects in vivo, not to mention the specificity of the PD0325901. The authors are trying to link the impaired bone recovery with macrophage polarization, which is not well supported by their data.
Comments on the Quality of English LanguageThere are many typos and grammar errors throughout the manuscript. The language needs to be improved.
Author Response
Reviewer 01
Comments and Suggestions for Authors
In this manuscript, the author treated mice with a MEK inhibitor after tooth extraction and assessed how this treatment affected the polarization of macrophage and bone regeneration. Several standard techniques in the field, such as micro-CT, histomorphometry, and birefringence were utilized here. However, considering the novelty and scientific rigor of current manuscript, I do not think it is ready for publication. Below are my detailed comments.
Answer: The authors are extremely thankful to the reviewers for the energy and time invested in reviewing this manuscript. We appreciate the detailed and constructive feedback that allowed us to improve the quality of manuscript.
- There are many typos and grammar errors throughout the manuscript, making the manuscript very challenging and distracting to read. Just name a few, the title, line 14, “theory mediate the cells migration …”, line 23, “resulted in an exacerbated a non-resolving”, line 71, “macrophages o predominate in the early stages”, and many more. The current manuscript is lengthy and wordy, which further increases the difficulty in comprehending the main message the authors want to convey.
Answer: As requested, we revised the manuscript to correct any grammatical errors, as well as to make the manuscript, as far as possible, more concise.
- I didn’t see the novelty of this manuscript, given previous studies have shown the regulation of macrophage polarization and bone repair by MEK signaling, such as PMID: 31801908, 38352056. What new knowledge does this current study bring?
Answer: We initially clarify that the references mentioned by the reviewer comprise studies with focus on lung, specifically on the involvement of MEK signaling in pulmonary alveoli under disease conditions, which completely differ from the study submitted, with focus on alveolar bone repair. In this context, while studies imply some theoretical roles for polarized macrophage subsets in bone repair, few mechanistic studies in fact were conducted to prove such roles in a cause-and-effect manner. In this study, we propose an alternative approach to elucidate the role of macrophage polarization for the alveolar bone repair process, which instead of using exogenous elements that can favor M2 polarization, we take advantage of compounds that interfere in cell signaling process to block the development of M2 phenotype. In conclusion, no previous studies used similar approaches to interfere in MEK1/2 along alveolar bone repair process, supporting the novelty of submitted study.
- Figure1, what conclusion are the authors trying to draw?
Answer: Figure 1 comprises a heat map (used in a standardized way by the group in other different works (Azevedo et al, 2021; Francisconi et al, 2023), which depicts the effects of the treatment with MEK1/2 inhibitor PD0325901 in the expression of several genes in different time points of the alveolar bone repair process. The target genes include growth factors (BMPs, TGFβ, VEGFs and FGFs), extracellular matrix markers (COL1a1, COL1a2, MMPs, TIMPs and Serpine), bone markers (RUNX2, DMP1, ALPL, PHEX, SOST, CTSK, RANKL, RANK and OPG), chemokines and their receptors (CCLs, CXCLs; CCRs and CXCRs), mesenchymal stem cell markers (CD34, 105, 106, 116, 146, OCT-4 and NANOG), cytokines (ILs, TNF and IFNG) and markers associated with macrophage polarization (iNOS, ARG, GAPDH, ACTb, HPRT-1 and FIZZ), being such factors representative of distinct stages of alveolar bone repair process. In this scenario, the overall conclusion is that the treatment with MEK1/2 inhibitor PD0325901 modulates the expression of macrophage polarization markers, and as discussed, that such modulation have consequences in other events of bone repair process, which include the expression of genes related to other events of repair.
- For the figures, what does the labels of a, b mean? The statistical results are very confusing. It is not clear how many biological repeats are performed in each experiment.
Answer: We clarify that the lowercase letters were used to identify statistical differences between different periods within the same group; where different letters represent statistical differences between the indicated periods, allowing the readers to evaluate possible differences in the kinetics of alveolar bone repair within each group. Such statistical notation has been used by our and numerous other research groups in order to minimize the use of multiple bars or more complex notation systems. To exemplify, if two-time points receive the ‘a’ notation, it means that there are no statistical difference between them; while a time point identified with ‘b’ notations statistically differs from the groups with ‘a’; if a group receive ‘ab’ notation there are no statistical difference when compared with groups noted with ‘a’ and ‘b’, but statistically differs from a group noted with ‘c’. Such clarification was included in the figure legends.
Regarding the number of biological repeats performed in each experiment, we clarify that the experimental N comprises 5 C57BL/6 wild type (WT) mice (N=5 group/time) - the same samples were used for microtomographic (μCT), histomorphometry, immunohistochemical and collagen birefringence analysis. In Real Time PCR Array analysis, we clarify that the experimental N comprises 4 C57BL/6 mice (N=4 group/time). As described and highlighted in yellow, in line 145 of materials and methods. Based on previous studies performed (Azevedo et al, 2021; Francisconi et al, 2023), to provide statistical power >90%. It is also important to consider that the isogenic strains, such as C57Bl/6, allow the use of a relatively low sample size to generate a proper statistical power, which is also in line with the ethical recommendations/guidelines for animal experimentation.
- For me, the representative images in fig. 4 are not as different as the authors claimed.
Answer: We clarify that while some parameters were identified as statistically different in the quantitative analysis, the images presented and the eventual comparison between them in a qualitative manner may be present the same degree of sensibility of the quantitative analysis. Please note that despite relevant differences identified by the statistical analysis of quantitative analysis, there are no massive/complete lack of repair in specific groups or time points, which would be more easily identified by the visual inspection of the figures presented.
- Is it Ly6c5b or ly6g5b (lines 309, 313)?
Answer: The marker is Ly6G5B. And as requested, we reviewed the manuscript and corrected our errors.
- Both “M1” and “M2” cells are reduced in the drug treatment group. It is unclear which subtype contributes to the delayed bone recovery.
Answer: The referee question highlights an important point. We initially clarify that the immunohistochemical analysis does not evidence significant variations in the counts of total macrophages, as well in M1 and M2 alleged macrophages, when control and MEK1/2i groups where compared. Conversely, the analysis of M1 and M2 markers by means of RealTimePCRarray suggest a modulation of M1/M2 balance, and as considered in the manuscript discussion, the blockade of MEK1/2 may interfere partially in the M2 phenotype, perhaps limiting the cells function instead of the acquisition of M2 phenotype along repair process. Therefore, the parallel analysis of both experiments is required for the analysis of macrophage polarization, which is also considered with additional details in the revised version of the discussion.
Regarding the possible involvement of M1 and M2 subsets in bone healing, as mentioned in the manuscript, while studies imply some theoretical roles for polarized macrophage subsets in bone repair, few mechanistic studies in fact were conducted to prove such roles in a cause-and-effect manner. In this study, we propose an alternative approach to elucidate the role of macrophage polarization for the alveolar bone repair process, which instead of using exogenous elements that can favor M2 polarization, we take advantage of compounds that interfere in cell signaling process to block the development of M2 phenotype. However, it is also mandatory to consider that MEK1/2 signaling is not exclusive of macrophages (please see also the answer to the next question in the sequence), and consequently, the effects of the use of PD0325901 in other cell types must be considered. Regarding the reviewer specific question (i.e. It is unclear which subtype {M1, M2} contributes to the delayed bone recovery), such aspect was also considered with more attention in the revised discussion section.
- Related to the above, MEK inhibition brings many biological effects in vivo, not to mention the specificity of the PD0325901. The authors are trying to link the impaired bone recovery with macrophage polarization, which is not well supported by their data.
Answer: We initially clarify and reinforce that previous studies demonstrate that PD0325901 is a specific inhibitor of MEK1/2. Therefore, considering the reviewer mention to ‘the specificity of the PD0325901’ and the subsequent comment regarding the attempt to ‘to link the impaired bone recovery with macrophage polarization, which is not well supported by their data’, we assume that the reviewer concern is related to the eventual specificity of the PD0325901 on macrophages. As mentioned in the above answer (to query #7), there are studies that evaluate the effect of the PD0325901 inhibitor on other cells as well, such as osteoclasts (Jiang et al, 2023; Kim et al, 2023), osteoblasts (El-hoss et al,2014), fibroblasts (Lee et al, 2024), as well in macrophages (He, et al, 2021, El-hoss, et al, 2014, De et al 2024) and neutrophils (De et al 2024). In the view of the reviewers’ concern, we added to the revised version of the discussion section a paragraph considering the potential action of PD0325901 in MEK1/2 in other cell types along alveolar bone repair process.
Reviewer 2 Report
Comments and Suggestions for Authors
The present study by Fonseca et al. investigates the critical role of the immune response on bone regeneration in alveolar sockets following tooth extraction. The authors specifically manipulate M2 macrophages through systemic administration of the MEK1/2 inhibitor to study its effects on cells and bone structure via cellular, morphological, and structural analyses. The reviewer would like to congratulate the authors for the quality and comprehensiveness of their work, which contributes to an in-depth understanding of healing mechanisms in tooth extraction sockets. However, a few issues need to be addressed:
-
Local vs. systemic effects of the MEK1/2 inhibitor: While the MEK1/2 inhibitor is expected to block macrophage polarization toward the M2 phenotype, qPCR analysis (e.g., upregulation of ARG) and immunohistochemistry (e.g., detection of CD206) suggest that this strategy was not entirely successful. The authors provide relevant discussion on this aspect. However, it remains unclear whether the systemic route achieved the intended local effect of the MEK1/2 inhibitor in the extraction socket. Please comment on this and add it to a new section on limitations in the Discussion.
-
Concentration of MEK1/2 inhibitor: How was the concentration selected? Did the authors perform a pilot study to determine an appropriate concentration for inhibiting M2 polarization in extraction sockets? In the Methods section, the authors refer to a study [reference 18] using this concentration to target cardiac tissue macrophages, which might require different concentrations than alveolar bone. Please address this and add it to the limitations in the Discussion.
-
Relevance and translational value: While the use of a MEK1/2 inhibitor emphasizes the crucial role of MEK1/2 in this context, its translational relevance appears questionable. Please comment and discuss this aspect.
-
Non-published data: The authors repeatedly refer to data on the drug diluent’s effects without presenting it. It is recommended to incorporate this data in the supplementary material.
-
Histomorphometry: The authors quantify osteoblasts and osteoclasts without specific stains or markers, relying solely on H&E staining. Please provide representative micrographs from each group to reflect the cell types identified with H&E.
-
qPCR methods: This section in the Methods does not match the Results in several aspects. For example, the Methods state that samples from all time points were pooled, while the Results provide data for each time point. Additionally, how was normalization performed? Was the delta-delta Ct method used?
- Graphs: please add all data plots in all bar graphs.
Author Response
Reviewer 02
Comments and Suggestions for Authors
The present study by Fonseca et al. investigates the critical role of the immune response on bone regeneration in alveolar sockets following tooth extraction. The authors specifically manipulate M2 macrophages through systemic administration of the MEK1/2 inhibitor to study its effects on cells and bone structure via cellular, morphological, and structural analyses. The reviewer would like to congratulate the authors for the quality and comprehensiveness of their work, which contributes to an in-depth understanding of healing mechanisms in tooth extraction sockets.
Answer: The authors are extremely grateful to the reviewers for the energy and time invested in reviewing this manuscript. We appreciate the detailed and constructive feedback that allowed us to improve the quality of the manuscript. All modifications made to meet reviewers' demands are highlighted in the revised version of the manuscript (highlighted in yellow).
However, a few issues need to be addressed:
Local vs. systemic effects of the MEK1/2 inhibitor: While the MEK1/2 inhibitor is expected to block macrophage polarization toward the M2 phenotype, qPCR analysis (e.g., upregulation of ARG) and immunohistochemistry (e.g., detection of CD206) suggest that this strategy was not entirely successful. The authors provide relevant discussion on this aspect. However, it remains unclear whether the systemic route achieved the intended local effect of the MEK1/2 inhibitor in the extraction socket. Please comment on this and add it to a new section on limitations in the Discussion.
Answer: The main limitation in carrying out a local approach would be to repeatedly inject this inhibitor in the repair site. While it is not possible to precisely determine the length of PD0325901 effect after injection, considering the relatively long duration of the experimental model used in this study, it was assumed that repeated injections would be required to assure the effect of the inhibitor in all the time points evaluated. Therefore, while it could be possible to locally deliver the inhibitor in the alveoli right after the tooth extraction, the subsequent development of a bone clot, followed over time by the development of a granulation tissue and its substitution by neoformed bone tissue, the effectiveness of the local injection would be questionable, since it would require the mechanical interference in the healing tissue, even considering the use of a fine needle, as possibly some limitations of the dispersion of the injected compound. In this scenario, the systemic injection was chosen for the experimental design used. In the view of the reviewer concern, we added to the discussion section a sentence/paragraph considering this aspect.
Concentration of MEK1/2 inhibitor: How was the concentration selected? Did the authors perform a pilot study to determine an appropriate concentration for inhibiting M2 polarization in extraction sockets? In the Methods section, the authors refer to a study [reference 18] using this concentration to target cardiac tissue macrophages, which might require different concentrations than alveolar bone. Please address this and add it to the limitations in the Discussion.
Answer: We clarify that a pilot study was carried out to observe the effects and select a possible dose for use, using 10mg/kg dose as the ‘target’ dose (based in previous literature reports) and concentrations 10x higher (i.e. 100mg/kg) and 10x lower (i.e. 1mg/kg) following the usual 10x variation dose/response curve pattern. However, it was observed that the 10x higher dose presented a high toxicity to animals, characterized by a high mortality rate. Considering that 10mg/kg and 1mg/kg doses resulted in similar mice response (i.e. lack of evident toxicity evidences), and considering that the 10mg/kg dose was already described in the literature (Sager et al., 2016) and confirmed in more recent studies (Sahu et al. 2021 and Jecrois et al. 2021), we choose to maintain the 10mg/kg dose administered daily intraperitoneally.
Relevance and translational value: While the use of a MEK1/2 inhibitor emphasizes the crucial role of MEK1/2 in this context, its translational relevance appears questionable. Please comment and discuss this aspect.
Answer: The reviewer highlights a very important point. As mentioned in the manuscript, while studies imply some theoretical roles for polarized macrophage subsets in bone repair, few mechanistic studies in fact were conducted to prove such roles in a cause-and-effect manner. In this study, we propose an alternative approach to elucidate the role of macrophage polarization for the alveolar bone repair process, which instead of using exogenous elements that can favor M2 polarization, we take advantage of compounds that interfere in cell signaling process to block the development of M2 phenotype. Therefore, despite the effectiveness of using PD0325901 compound to demonstrate the effect of MEK1/2 inhibition on bone healing outcome in a cause-and-effect, and the constructive role of M2 macrophages required for proper bone healing, we agree with the referee and such strategy would not make sense from the translational viewpoint. However, the results presented here can provide the basis for future studies with translational potential, perhaps focused in improving – and not inhibiting – MEK1/2 signaling. Such aspects were considered in the revised version of the manuscript.
Non-published data: The authors repeatedly refer to data on the drug diluent’s effects without presenting it. It is recommended to incorporate this data in the supplementary material.
Answer: As requested, we added data about the diluent in the supplementary material.
Histomorphometry: The authors quantify osteoblasts and osteoclasts without specific stains or markers, relying solely on H&E staining. Please provide representative micrographs from each group to reflect the cell types identified with H&E.
Answer: As requested, we added microphotographs of the different cell types to the supplementary material.
qPCR methods: This section in the Methods does not match the Results in several aspects. For example, the Methods state that samples from all time points were pooled, while the Results provide data for each time point. Additionally, how was normalization performed? Was the delta-delta Ct method used?
Answer: As requested, the qPCR methods and results description were improved to clearly present the details required. We revised the incorrect statement that ‘samples from all time points were pooled’ (such method was used in a previous study, and the description was not properly revised to match with the exact method used in this study).
We clarify that data analysis was performed with RT2 profiler PCR Array Data Analysis online software, which comprise an initial normalization in relation to three constitutive genes (GAPDH, ACTB, Hprt1), and a subsequent normalization using regular maxillary alveolar bone as the normalizing control. Finally, the statistical significance of the experiment involving the PCR Array was evaluated by the Mann-Whitney test, and the values tested for correction by the Benjamini—Hochberg Procedure. Such information was included in the revised version of the manuscript.
Graphs: please add all data plots in all bar graphs.
Answer: As requested, we added all data plots to the bar graphs.
Reviewer 3 Report
Comments and Suggestions for Authors
Dear Author,
Fonseca et al. present a study examining the role of MEK1/2 signaling in the activity of macrophages during bone repair. In a mouse tooth extraction model, the MEK1/2 inhibitor PD0325901 was administered. PCR array analysis was performed on RNA extracted from the alveolar bone area at 0, 3, 7 and 14 days. MEK1/2 inhibition increased BMP expression but reduced VEGF. It also modulated chemokines (mostly upwards) bone and MSC markers. In immunohistochemical confirmation of these results, cell counting was performed for Ly6C5B/Gr1+ (granulocytes), CD3-e+ (T lymphocytes), CD206+ (M2), CD16&CD32+ (M1) and F4/80+ (M0). The pathology profiling followed accumulation and disappearance of the various cell types over 14 days, with MEK1/2 inhibition generally reducing the number of all infiltrating cell types. In 3D uCT analysis, bone volume restoration was reduced by MEK1/2 inhibition, as were several trabecular bone formation parameters. In further histological analysis of the components of the repairing extraction site most differences were noted on day 14, where MEK1/2 inhibition increased blood vessels but reduced bone matrix, osteoblasts and osteoclasts. Finally, in birefringent collagen quantification, MEK1/2 inhibition significantly reduced collagen deposition in the extraction site thought there were no major changes in the maturation phase the fibers.
This is a well written paper and the techniques employed are good and appear competently performed. The histology is of high quality.
Major observations
------------------
1. I would like to see an in vitro proof of PD0325901 effects on macrophages polarization.
2. I had trouble understanding the significance markers on the graphs. e.g. Figure 3 "*(p<0.05) indicates statistically significant differences between control versus MEK1/2i group." Can these be presented in a different fashion for clarity?
3. It is considered insufficient to rely on one marker for macrophage phenotyping, as there is much overlap and heterogeneity between M0/M1/M2.
4. Were there any observable health differences between control and MEK1/2 inhibition animals? This needs to be detailed as, of course, any general condition changes would also impact bone repair.
Minor observations
------------------
1. Perhaps there is a paste issue at the 2.6 leading to a line spacing change.
Regards
Author Response
Reviewer 03
Comments and Suggestions for Authors
Fonseca et al. present a study examining the role of MEK1/2 signaling in the activity of macrophages during bone repair. In a mouse tooth extraction model, the MEK1/2 inhibitor PD0325901 was administered. PCR array analysis was performed on RNA extracted from the alveolar bone area at 0, 3, 7 and 14 days. MEK1/2 inhibition increased BMP expression but reduced VEGF. It also modulated chemokines (mostly upwards) bone and MSC markers. In immunohistochemical confirmation of these results, cell counting was performed for Ly6C5B/Gr1+ (granulocytes), CD3-e+ (T lymphocytes), CD206+ (M2), CD16&CD32+ (M1) and F4/80+ (M0). The pathology profiling followed accumulation and disappearance of the various cell types over 14 days, with MEK1/2 inhibition generally reducing the number of all infiltrating cell types. In 3D uCT analysis, bone volume restoration was reduced by MEK1/2 inhibition, as were several trabecular bone formation parameters. In further histological analysis of the components of the repairing extraction site most differences were noted on day 14, where MEK1/2 inhibition increased blood vessels but reduced bone matrix, osteoblasts and osteoclasts. Finally, in birefringent collagen quantification, MEK1/2 inhibition significantly reduced collagen deposition in the extraction site thought there were no major changes in the maturation phase the fibers.
Answer: The authors are extremely thankful to the reviewers for the energy and time invested in reviewing this manuscript. We appreciate the detailed and constructive feedback that allowed us to improve the quality of manuscript.
This is a well written paper and the techniques employed are good and appear competently performed. The histology is of high quality.
Major Comments:
- I would like to see an in vitro proof of PD0325901 effects on macrophages polarization.
Answer: We agree with the reviewer viewpoint regarding the importance of in vitro analysis of PD0325901 effects on macrophages polarization. We initially clarify that previous studies (Long et al, 2017; El-hoss, et al, 2014, De et al 2024), performed in vitro investigation of PD0325901 effects on macrophages polarization, including a relatively recent investigation performed a broad characterization of macrophage polarization mechanisms and identification of M2-type polarization inhibitors (He, et al, 2021), which included PD0325901, described in the study as one of the most potent inhibitors of IL-4-induced macrophage polarization in the screens were multiple highly selective MEK inhibitors were tested. Additionally, a previous study demonstrates that PD0325901 significantly increased expression of IL-4/IL-13 (M2) responsive genes in murine bone marrow-derived and alveolar macrophages.
In addition to the previous support from the literature, we choose to not perform any in vitro analysis of the putative PD0325901 effects on macrophages polarization in the view of the unclear nature of M2 polarizing factors in healing sites. Noteworthy, in vitro experiments focused on M2 polarization process are essentially based in the use of IL-4 as the prototypic M2 polarizing factor. However, as mentioned in the manuscript, Th2 cytokines, such as IL-4 and IL-13, are not usually detected in bone injury/repair sites. Therefore, while it is possible to suggest that other cytokines, such as VIP and PACAP, could mediate M2 polarization in healing sites, the uncertainness of this suggestion does not support strong statements regarding the real factors governing M2 polarization in healing sites.
Therefore, the use of IL-4 based experiments would replicate previous data, and the use of other factors would require broader experiments beyond the scope of this study.
- I had trouble understanding the significance markers on the graphs. e.g. Figure 3 "*(p<0.05) indicates statistically significant differences between control versus MEK1/2i group." Can these be presented in a different fashion for clarity?
Answer: We assume that the figure 3 was used as example of the lack of clarity, and indeed, since specifically in Figure 3 there is no statistical difference, therefore there is no (*) despite the the figure caption contained the following description (*(p<0.05) indicates statistically significant differences between the MEK1/2i versus control group), which may be the main cause of the lack of clarity. Figure 3 legend was correct to avoid the issue. To improve the clarity and to allow a better visualization, the * (asterisk) was placed above the letters in the experimental group (MEK1/2), always indicating a statistical difference when comparing the control group and experimental group.
- It is considered insufficient to rely on one marker for macrophage phenotyping, as there is much overlap and heterogeneity between M0/M1/M2.
Answer: We agree with the reviewer’s viewpoint that additional data would provide a more robust support to the finding. However, co-staining attempts were not considered satisfactory from the technical aspect and consequently were not added to the manuscript. Also, our group tried to perform flow cytometry-based cellular characterization, as previously performed with periodontal tissues samples. However, due to the requirement of additional tissue processing due the features of bone (and forming bone) tissue (the samples require stronger enzymatic and mechanical processing when compared with periodontal tissues to allow the ‘cell release’ from matrix), it was not possible to recover cells in number and quality (specially viability) enough to the flow cytometer analysis. In this context, we consider that the use a broad panel containing macrophage markers in the RealTimePCRarray analysis provide some additional support to the interpretation of the global macrophage response scenario. This complementary nature of molecular analysis obviously does not overcome the limitations of the limited IHQ analysis with single markers, and this limitation was considered in the revised discussion section.
- 4. Were there any observable health differences between control and MEK1/2 inhibition animals? This needs to be detailed as, of course, any general condition changes would also impact bone repair.
Answer: All animals that might show any changes were immediately excluded from the sample. By convention, the daily administration of the MEK1/2 inhibitor allowed us to observe the animals' behavior. Regarding the dose of the inhibitor, we carried out a pilot study to observe the effects and select a possible dose for use. Concentrations 10x higher and 10x lower following the dose/response curve pattern. The 10x higher dose, however, presented high toxicity to animals with a high mortality rate, so we chose to maintain the 10mg/kg dose administered intraperitoneally with daily receipt of the medication as already described in the literature (Sager et al., 2016) and confirmed in studies more recent (Sahu et al. 2021 and Jecrois et al. 2021). Therefore, we assume that no systemic changes were considered significant to report.
Minor Comments:
- Perhaps there is a paste issue at the 2.6 leading to a line spacing change.
Answer: As requested, we revised the manuscript and corrected the line spacing 2.6
Round 2
Reviewer 1 Report
Comments and Suggestions for Authors
I appreciate the authors' efforts to address my concerns and response to my comments. However, most of my initial comments, both scientifically and technically, are not addressed well and the authors failed to resolve my concerns. I don’t think the manuscript at its current form is qualified to be published.
1. Some of the responses are very confusing. For instance, #4 " we clarify that the experimental N comprises 5 C57BL/6 wild type (WT) mice (N=5 group/time)". Another example is that in the response letter, "To exemplify, if two-time points receive the ‘a’ notation, it means that there are no statistical difference between them”, while in the legends for figures 3, 5, 7, 9, “Lowercase letters represent a statistically significant difference (p<0.05) between different periods within the same group.”
2. For the heatmap in fig 1, the authors unnecessarily spent large paragraph describing the data, again, what is their conclusion? The authors should incorporate their response to #3 into manuscript.
3. The language should be improved and the text be more concise. It is very challenging to read and comprehend current version.
Comments on the Quality of English Language
The language should be improved and the text be more concise. It is very challenging to read and comprehend current version.
Author Response
Comments and Suggestions for Authors
I appreciate the authors' efforts to address my concerns and response to my comments. However, most of my initial comments, both scientifically and technically, are not addressed well and the authors failed to resolve my concerns. I don’t think the manuscript at its current form is qualified to be published.
- Some of the responses are very confusing. For instance, #4 " we clarify that the experimental N comprises 5 C57BL/6 wild type (WT) mice (N=5 group/time)". Another example is that in the response letter, "To exemplify, if two-time points receive the ‘a’ notation, it means that there are no statistical difference between them”, while in the legends for figures 3, 5, 7, 9, “Lowercase letters represent a statistically significant difference (p<0.05) between different periods within the same group.”
Answer: We clarify that the experimental N comprise 5 mice for each time for each group; such information was originally displayed in the manuscript as "N=5 group/time". For experimental logistics reason, the only experiment with a different size sample is the RealTimePCRarray, in which 4 mice for each time for each group were used, such information was originally displayed in the manuscript as "N=4 group/time". For clarity reasons, the M&M section was revised as requested.
The revised section now reads: The experimental sample size comprise 5 mice for each time point for each group (N=5 group/time point) for microtomographic (μCT), and then prepared for histomor-phometry, immunohistochemical and collagen birefringence analysis, except for RealTimePCRarray in which 4 mice for each time for each group were used for experi-mental logistics reason, (N=4 group/time point). In the end of the experimental periods, animals were euthanized with an excessive dose of anesthetic and the maxillae were collected.
Regarding statistical differences, we use different lowercase letters to refer to statistical analysis between periods of the same group. The two sentences mentioned are grammatically different but its content is not conflicting or dissonant. For clarity, the figures legends were modified and now reads: `Different lowercase letters represent a statistically significant difference (p<0.05) between different time points within the same group; time points within the same group presenting the same letter are not statistically different`.
- For the heatmap in fig 1, the authors unnecessarily spent large paragraph describing the data, again, what is their conclusion? The authors should incorporate their response to #3 into manuscript.
Answer: While we agree with the reviewer that gene expression data description results in a large paragraph, such description is necessary and the relatively large paragraph is derived from the large volume of data comprising the figure. Therefore, the description is considered necessary, and it is presented in a more concise way possible considering the large data volume context. The overall conclusion is that MEK1/2 inhibition results in modulation of the gene expression profile, and such modulation is explored in details in the discussion section.
- The language should be improved and the text be more concise. It is very challenging to read and comprehend current version.
Answer: Considering the reviewers' comments, we read the manuscript again with the aim of improving its writing.